# No Place Called Home. The Banishment of 'Foreign Criminals' in the Public Interest: A Wrong without Redress

**Helen O'Nions** 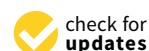

Nottingham Law School, Nottingham Trent University, Nottingham NG1 4BU, UK; helen.onions@ntu.ac.uk

**Abstract:** This article examines the legal and ethical rationale for the deportation of 'foreign criminals' who have established their homes in the United Kingdom. It argues that provisions relating to automatic deportation constitute a second punishment that can be more accurately described as banishment. The human rights of those defined as 'foreign criminals' have been reduced to privileges that are easily withdrawn with reference to the ill-defined public interest. The ability to challenge deportation is then compromised by a non-suspensive appeal process that deliberately undermines the right to an effective remedy whilst further damaging private and family life. With reference to social membership and domicile theories of belonging, it is suggested that those who have made their lives in the UK and established their place and domicile here should be regarded as unconditional members of civil society. As such, they are entitled to equality of treatment in the criminal justice system and should be immune from punitive 'crimmigration' measures.

**Keywords:** deportation; citizenship; foreign criminals; family life; human rights; appeals; 'hostile environment'

## 1. Introduction

In 2012 the then Home Secretary Theresa May announced the introduction of a 'really hostile environment' for 'illegal immigrants'. Absent of any robust impact assessment, a series of legislative and policy measures followed, the consequences of which continue to be felt by all who have migrated to the UK and many who regard it as their home (Williams 2020). This article focusses on the position of established residents who have engaged in criminal activity and face deportation. It is argued that removal is akin to banishment and that it is ethically wrong with reference to normative understandings of belonging and membership. Furthermore, it is legally wrong with reference to fundamental human rights norms that are decoupled from formal citizenship status.

There are strong ethical arguments presented in theories of social membership that support a right of residence for those considered to be 'citizens in the making' (Miller 2008, p. 195). Yet those defined legally as 'foreign criminals' present a challenge to membership theories as the foundations of this right are typically predicated on good behaviour, measured for example through the strength of social and cultural ties or positive contributions made to the host society. Indeed, the offender's criminal history can be presented as evidence that no such ties exist, undermining an ethical argument against expulsion. The argument in this paper is that there is no meaningful qualitative difference between citizenship and permanent residence for the purpose of ascribing membership. Membership is a question of fact, existing irrespective of criminal behaviour in much the same way as it exists for citizens. To refuse membership rights results in civic marginalization (Owen 2013). This argument is grounded in both social membership and domicile theories of non-deportability advanced by Carens, Moore and Birnie, that support an unconditional right of residence irrespective of formal citizenship status.

Whilst it may be possible to have membership of more than one national community this is far from typical. Recent deportations suggest that those with indefinite leave consider themselves to be British and rarely have significant ties to their country of nationality, in some cases the deportee has spent their entire life in the UK. British citizenship is not necessarily acquired by place of birth so it is conceivable that some permanent residents would not even be aware they lacked citizenship. Post sentence detention and deportation in these situations constitutes a second punishment, amounting to enforced exile or banishment.

The percentage of foreign nationals serving prison sentences is comparable to the percentage of foreign nationals living in the UK generally (Sturge 2019) yet the 'foreign criminal' has become 'doubly damned' as an ungrateful, 'bad' migrant, whose very existence threatens the community of value (Griffiths 2017). The label is enduring, reducing the individual to a moment in time that will henceforth define every aspect of their identity. The introduction of automatic deportation in the UK Borders Act 2007 essentially confirms this position. The offender's rights, and those of their families, become privileges that have been abused. This can be seen clearly in the proportionality assessments of decision-makers which, it is argued, often appears cursory.

Deportation of offenders is justified in the legislation by reference to the public interest. The implication being that it is necessary for public protection and the prevention of crime. Foreign criminals certainly elicit little public sympathy, although this may in part be attributable to the way that deportations are framed in public discourse. It is hard to argue against the view that very serious offenders, such as murderers and drug traffickers, present a threat to public safety. The inability of previous Home Secretaries to remove foreign nationals following completion of their sentence has attracted a great deal of public condemnation and led to the resignation of Home Secretary Charles Clarke in 2006. Such a failure appears to undermine the first duty of the government to keep citizens safe and the country secure.

Recent amendments to the UK's immigration rules and the introduction of s117C of the Nationality, Immigration and Asylum Act 2002 (hereafter 'NIAA') pre-load the decision-makers assessment of proportionality in favour of expulsion where the individual was sentenced to twelve months in prison, but make some allowances for arguments based on both private and family life (reflecting the UK's obligations under Article 8 of the European Convention on Human rights). Article 8 is given effect through the Human Rights Act 1998 but it is a qualified right and the state has been afforded a margin of appreciation in its assessment of the public interest. Yet the public interest is presented as a given in deportation legislation and therefore receives little scrutiny, as evidenced by the snapshot of cases presented below. Zedner notes how criminals are typically characterised in public discourse as enemies in a way that ignores research on the normality of offending (Zedner 2010, p. 390). All offending is treated as equally dangerous and there is no willingness to look behind the crime. 'Foreign criminals' (and their families) cease to be members of the public when their enemy status is consolidated by the absence of formal citizenship.

The steady devaluation of Article 8 rights observed by (Griffiths 2017, p. 533) has been accompanied by a corresponding devaluation in procedural rights including the right to challenge a deportation order. One of May's flagship 'hostile environment' policies was the 'deport now, appeal later' provision enacted by the Immigration Act 2014. May had previously expressed frustration at the judiciary for ignoring the will of Parliament when "putting the law on the side of foreign criminals instead of the public" (House of Commons 2013, col. 156; O'Nions 2020). Legal representatives were also accused of 'cashing in' and the appeals system was characterized as an 'abuse of Article 8' (House of Commons 2013, col. 158; O'Nions 2020). However, the Supreme Court in *Kiarie and Byndloss* ruled that the provisions undermined the right to an effective appeal, both in terms of its substance (creating further damage to the applicant's private and family ties) and in terms of the process which needs to

be effective.[1] It is now accepted that for an effective appeal, deportees may need to be returned to the UK after several months to present their evidence and face cross-examination.[2] Yet statistics suggest that very few deportees will attempt to challenge their removals once overseas, the reasons for this are unclear but it is likely that cost and accessibility are significant factors.

Given the argument that deportation of those permanent residents perceived to present a danger to an ill-defined public interest is both ethically and legally problematic, there is a further need to reflect on the underlying rationale of conditional membership. The post-Brexit landscape reveals a country deeply divided economically, socially and culturally. To a large extent deportation shores up the foundations of the community of value by overtly signalling that certain behaviour is unwanted whilst confirming that the immigration system is not a 'soft touch' (Walters 2002, p. 286). However, it paradoxically contributes to the fragility of membership by emphasising the enduring 'foreignness 'of some members of that community and reducing their fundamental human rights to precarious privileges. 'Hostile environment' policies such as the right to rent scheme which requires private landlords to check the immigration status of their tenants, have arguably legitimized discrimination against those perceived to be foreign and further contributed to this fragility (Independent Chief Inspector of Borders and Immigration 2018).[3]

It is important to begin by contextualizing the ethical and legal arguments against deportation with reference to the most recent deportations and the legislative framework. This also provides an insight into the way that the 'hostile environment' has constructed certain types of foreigner in public discourse. A consideration of the ethical arguments against deportation will follow in an attempt to ground a theory of unconditional membership for permanent residents. Finally, the paper addresses the human rights principles that are relevant in this context. It will be argued that the rights of 'foreign criminals' and their families are too easily reduced to privileges when balanced against the ill-defined public interest.

## 2. Setting the Context

Section 32 of the UK Borders Act 2007 provides for automatic deportation of those termed 'foreign criminals' which becomes effective when the individual receives a prison sentence of at least 12 months. Further, it allows the Home Secretary to specify offences that are deemed to be 'particularly serious' where any sentence can constitute grounds for deportation. Regulations introduced pursuant to this section were deemed ultra vires on the grounds of irrationality in *EN(Serbia)* [2009] as they specified offences, such as theft and criminal damage, which were not necessarily 'serious'.[4] This has resulted in a rebuttable presumption of dangerousness.

Legislation introduced in 2014 pursuant to the 'hostile environment' established that in the case of 'foreign criminals' deportation was in the public interest, removing any need for the deporting authority to pay specific attention to the nature of the offence, the history of offending and current assessment of risk (s117C Nationality Immigration and Asylum Act 2002). A 'foreign criminal' may be detained immediately following the end of their sentence or many months later and there is no automatic bail hearing. A very recent Court of Appeal decision found that there was a real risk of detainees being removed by the Home Office before they had an opportunity to challenge an adverse decision before a court. The notice period required for deportation did not provide sufficient opportunities for the deportee to access legal advice before the removal window became operational.[5]

---

1   Kiarie and Byndloss v Secretary of State for the Home Dept. [2017] UKSC 42.
2   AJ (s 94B: Kiarie and Byndloss questions) Nigeria [2018] UKUT 00115 (IAC).
3   R (Joint Council for the Welfare of Immigrants) v Secretary of State for the Home Dept. [2019] EWHC 452 (Admin).
4   EN (Serbia) v Secretary of State for the Home Dept.; Secretary of State for the Home Dept. v KC (South Africa) [2009] EWCA Civ 630.
5   R (On application of FB) v Secretary of State for Home Dept. [2020] EWCA Civ 1338.

'Foreign criminals' are typically detained for longer periods than other immigration detainees, on average over four months (Shaw 2018, para. 4.98). The absence of a time-limit, the challenges of securing legal representation (legal aid is not routinely available in immigration cases) and anxiety over the ever-present possibility of expulsion leads many to report a deterioration in their mental health (Chief Inspector of Prisons 2017; Borril and Taylor 2009). The Home Office do not ordinarily provide specific deportation dates so they can reduce the possibility of late legal challenges. This undoubtedly adds to the anxiety of detainees. One of the foreign criminals interviewed during Lord Shaw's investigation of immigration detention had been in the UK since birth. He had committed a gang-related offence as a teenager and was awaiting deportation to Nigeria, a country that had so far refused to accept him. He had been detained at Campsfield House for more than fourteen months at the time of the report, notwithstanding a review recommending his release. His Home Office file stated that he was not socially or culturally integrated in the UK due to his involvement in crime (Shaw 2018, para. 4.98).

Lawyers report difficulties in accessing clients in detention whereas detainee advocacy groups argue that the quality of legal advice given to detainees is poor (Bail for Immigration Detainees 2020a). There has been a proliferation of providers under the current duty detention scheme but this has meant that many are inexperienced and do not have the time to build up expertise. A survey undertaken by BID in 2018 had found that only half of detainees had a legal representative and whilst this had improved somewhat in 2019, 40% of immigration detainees still lacked representation. For those detained in prison following the end of their sentence, only 15% had received legal advice from an immigration solicitor (Bail for Immigration Detainees 2019).

The recent deportation flights from the UK to Jamaica illustrate the inherent contradictions in deportation policy. The testimonies of those detained pending deportation can be contrasted to the official narrative and serve as a reminder of both the normality of offending and the personal impact of expulsion, described by Griffiths as 'life-altering violence' (Griffiths 2017, p. 538).

On the 6 February 2019, twenty-nine 'foreign criminals' were deported to Jamaica whilst more than 50 others were returned to detention following last minute appeals (Gentleman et al. 2019). A second flight carrying seventeen 'foreign criminals' departed in February 2020. At least twenty-five others were prevented from leaving due to a court order which found that faulty phone masts had prevented them accessing their lawyers whilst detained (BBC News 2020a). Home Office minister Kevin Foster insisted that there were no 'British nationals' onboard the flight and emphasised the dangerousness of all those on board (Honeycombe-Foster 2020). It was repeatedly asserted that all were violent offenders (BBC News 2020a; Daily Mail 2020). The Home Office press release stated:

> "Today 17 serious foreign criminals were deported from the UK. They were convicted of rape, violent crimes and drug offences and had a combined sentence length of 75 years, as well as a life sentence. We make no apology whatsoever for seeking to remove dangerous foreign criminals". (BBC News 2020a)

In the face of such obvious dangerousness it is understandable that removal received broad public support but the individual stories behind the headlines were more complicated. Several of the deportees, their representatives and partners were subsequently interviewed by journalists from the BBC, *The Guardian* and *The Independent* newspapers. These testimonies challenge official accounts and provide a valuable insight into the experience of post-sentence detention and deportation (Merrick 2020).

One notable feature in several testimonies is the period of delay between release from prison and the decision to re-detain, calling into question the public interest justification. David Lammy MP highlighted the case of Tayjay who was twenty-four years old when detained, four years after his release from prison. Tayjay had been in the UK since the age of five and all his family live here. Having been groomed to participate in domestic drug trafficking (known as 'county lines') as a minor he was sentenced to fifteen months in prison at the age of nineteen. There was no suggestion of any further criminality following his release from prison (Lammy 2020). BBC Newsnight featured the story of Rayan Crawford who came to the UK aged 12, twenty years ago (Clayton 2020). He has a history

of minor offending and served a twelve-month sentence for burglary in 2017. He was detained two years after his release from prison pending deportation. Rayan has a partner of fourteen years and two children who are British citizens. He also suffers from a degenerative bone disease and inflammatory arthritis which require regular medication. A tribunal hearing found that although the deportation would be very difficult for the family it would not meet the legal threshold of being 'unduly harsh'. His family were unable to afford the £2000 needed for a judicial review and he was reportedly deported in February without his medication.

The public interest justification is also arguably weak where the level of offending is below the statutory threshold of twelve months imprisonment. The charity Bail for Immigration Detainees highlighted the case of EB who was on the February flight. Having come to the UK aged eight he now has a British partner and four children all of whom are British citizens. While EB has committed low-level offences in the UK he has never received a custodial sentence amounting to 12 months which would make him automatically liable to deportation. He was assessed to be at low risk of reoffending by the probation service (Bail for Immigration Detainees 2020b). Reshawn Davis was removed from the deportation flight in February. Although he had been convicted of robbery ten years ago this was under a test for joint enterprise that the Supreme Court subsequently found to be incorrect. Despite being subject to automatic deportation powers, Reshawn served only two months in prison and has not committed any further crimes. He has lived in the UK since the age of 11 and has a British partner and five-month old baby (Bulman 2020). His solicitor informed *The Independent* that despite living with his partner and daughter the Home Office rejected his argument that he had a genuine and subsisting relationship with them. Further, they found that it would not be unduly harsh on the family for Reshawn to be deported. Davis was interviewed two weeks later whilst still in detention. At this point he had received no information about his case and did not know whether he would be released or deported.

The BBC sympathetically highlighted the case of Rupert Smith dubbed a violent 'thug' by the tabloid media. Rupert came to the UK age 11 with his parents and all his family live here (Murphy-Bates and Law 2020). After finishing school, he spent four years at college when he committed his only offence of Actual Bodily Harm in retaliation for a sexual offence committed against a member of his family. Rupert described detention as 'like being on death row' (Taylor 2019. He arrived in Jamaica with only the clothes he was wearing and was taken to an army barracks where a volunteer offered him temporary accommodation.

MP Shabana Mahmood revealed that one of her constituents was removed from the most recent flight. Having fought in the army on two tours of Afghanistan he now suffers from post-traumatic stress disorder and has been diagnosed with bi-polar affective disorder. The injustice of seeking to deport a man who has risked his life defending the security of the country is not lost on Mahmood: "he's served this country; he hasn't had help for the PTSD he picked up as a serving soldier for our country . . . it really goes to the heart of our notions of what it is to be a citizen" (BBC News 2020a).

It is evident from this snapshot of cases that the Government's repeated assertion that all those on the deportation flight were 'rapists and violent criminals', simply cannot be supported. Many had committed their offences when teenagers, were not repeat offenders and were arguably unlikely to reoffend (BBC News 2020b). In these cases, arguments concerning the public interest ring hollow (Shaw 2018, para. 4.99).

Griffiths argues that "offenders in the UK today . . . are punished harder and denied redemption when they are non-citizens, casting them as more seriously and indelibly criminal than their British counterparts" (Griffiths 2017, p. 531). Policing practices which target young black men and the prevalence of gang culture in some of the UK's biggest cities feed into the statistics on deportation. De Noronha contends that racialized policing and discrimination inevitably mean that a disproportionate number of young black men are incarcerated, and this is played out in the decisions to expel (De Noronha 2019, p. 2425).

This is most apparent when considering the partnership between police and immigration officials known as Operation Nexus. Nexus was piloted in London in 2012, the same year as Theresa May announced that Britain would become a 'really hostile environment' for illegal immigrants. The data sharing at the heart of Nexus has now been rolled out in many areas of the UK. It is framed as targeting 'High Harm' foreign national offenders but it has become apparent that those with spent and petty convictions, as well as 'non-convictions', such as police encounters, acquittals and withdrawn charges are being targeted (Home Office 2017; De Noronha 2019). Between 2012 and 2015, 3000 FNOs were removed under Nexus, with the figure expected to increase after Brexit as European Union citizens become subject to British immigration rules. Nexus is grounded in an unsubstantiated assumption that migrant communities are more likely to engage in crime (Zedner 2010, p. 386), with the result that at a case for deportation can rest on:

> "a medley of allegations, associations, unproven assertions, hearsay, anonymous evidence, the behaviour of the appellant's friends and circumstantial evidence, none of which would usually be admissible in a criminal court". (Griffiths 2017, p. 533)

The numerous procedural and evidential requirements safeguarding the rights of criminal defendants are not required by the administrative process of removal. Griffiths details several cases marked for deportation following a Nexus investigation including a 20-year-old man who had been in the UK since the age of 5. His longest sentence was eight weeks for carrying a knife which he attributed to the need to protect himself after being stabbed three times (Griffiths 2017, p. 537). Another man interviewed by Griffiths had overstayed and had experienced prolonged periods of alcoholism and destitution. He had been detained for 5 months pending deportation after stealing a piece of steak at new year. These examples illustrate how the political construction of the 'foreign criminal' is inextricably linked to the creation of the 'hostile environment'. The use of speculation and assertion to justify forced exile of non-citizens can hardly be said to be in the public interest. The legality of Operation Nexus will soon be assessed by the Supreme Court after the Court of Appeal granted leave to appeal their ruling confirming its lawfulness.[6]

Prisons and removal centres are similarly focussed on the end game of exile. In her study of immigration removal centres, Bosworth observes how very little support is offered to 'foreign criminals' in terms of assisting reintegration and rehabilitation because it is understood that their imprisonment is "geared to one aim: deportation" (Bosworth 2011, p. 586; Stumpf 2006, p. 408). Prison overcrowding now affects 62% of prisons and remand facilities are particularly affected, leaving many confined to cells following their release date (Sturge 2019).

Hasselberg interviewed deportees and their families who were on bail pending deportation following a period of immigration detention (Hasselberg 2014, p. 471). All participants regarded themselves as settled in the UK despite the heterogeneity of their backgrounds. Their impression of surveillance strategies such as reporting are conceptualised as a form of coercive action which compels them to depart by constraining their lives into an ever-decreasing space of confinement. In 2016 the Government's plan to tag all released foreign criminals and subject them to a curfew was deemed unlawful.[7] Many of Hasselberg's participants regarded the condition of deportability and bail as a trap to make them more likely to fall into further criminal behaviour. The characterisation of powers such as detention and expulsion as administrative procedures is directly contradicted by the experiences of 'foreign criminals' who regard them as additional, arbitrary punishments (Dow 2007; Aas and Bosworth 2013).

---

6  Centre for Advice on Individual Rights in Europe) v Secretary of State for the Home Dept. and Commissioner of Police for the Metropolis [2018] EWCA Civ 2837.
7  R (On the application of Abdiweli Gedi) v Secretary of State for Home Dept. [2016] EWCA Civ 409.

### 3. From British Subject to 'Foreign Criminal'

Walzer observes how that the study of distributive relationships within the political community always begs the prior question of how that community was constituted and maintained in the first instance (Walzer 1983, p. 30). Any discussion on the subject of removals and membership must be viewed in the context of the UK's colonial history, recently played out in the Windrush affair (Williams 2020).

Prior to the Commonwealth Immigrants Acts in 1962 and 1968 citizens of the UK and colonies could freely enter and reside in the UK without restrictions, based on their implied allegiance to the Queen. There were certainly voices arguing that deportation should be applied to certain groups of foreign criminals, typically those from the 'new' commonwealth (as distinguished from the old, largely white, commonwealth). Yet calls to extend deportation powers were initially resisted as subjects of the empire were considered to be full members of the British 'community of value' (Gibney 2013, p. 225). By 1971 this position had changed with the introduction of the Immigration Act, but the power to deport was rarely used until the enactment of the UK Borders Act and its particular construction of the 'foreign criminal' (s32(1) UK Borders Act 2007).

As De Noronha argues, colonial histories and global inequalities are often obscured in immigration discourses, enabling important questions about racism to be pushed aside (De Noronha 2019, p. 2416). This can be very clearly seen with the Windrush generation who left their homes in British colonies at the invitation of the British Government from 1948 onwards. Having grown up in the UK, paid taxes and built their lives here, the Windrush generation and their children regarded themselves as British. The Immigration Act 1971 provided that those ordinarily resident for five years at the date of commencement were entitled to citizenship. Yet in 2012, shortly after the introduction of the 'hostile environment' it became apparent that the Home Office were disputing their membership. As a result, 83 people were unlawfully deported, others lost their jobs, and some were refused urgent medical treatment (Williams 2020). The Home Office had destroyed the landing cards that could have proved entitlement to citizenship and without such proof had treated everyone as a foreign national, notwithstanding national insurance and other official records.

Some of the recent expulsions concern relatives of the Windrush generation. Most identify exclusively as British and struggle to comprehend the sudden realisation of the precarity of their membership (Hasselberg 2014; Grell 2020). Whilst ministers are keen to distinguish their deportations from the Windrush cases, there are notable comparisons. It is no accident that the publication of the review into Windrush was delayed for over a year until the departure of the most recent flight to Jamaica. The 'Lessons Learned' review rebuts claims that Windrush was both unforeseen and unavoidable, placing the blame on a Home Office culture which is ignorant of history and defends, deflects and dismisses criticism (Williams 2020). The 'hostile environment' along with the Home Office's well-documented culture of disbelief, lies at the root of the Windrush affair. The Home Office, driven by removal targets, ignored the sensitivities of individual cases including lifetimes of lawful residence, extensive family ties and contributions to British society. The public outrage over the Windrush affair centred on the denial of membership and the hardship that resulted from expulsion. Whilst the 'foreign criminal' inevitably elicits less public sympathy the same central arguments apply. Indeed, the Government initially defended many of the 83 Windrush deportations on the grounds of criminality (Gentleman 2018) and subsequently excluded these cases from the 'Lessons Learned' review.

'Lessons Learned' criticises the inflammatory rhetoric used by ministers, particularly when it comes to the subject of 'foreign criminals'. Similar concerns were raised five years ago by senior civil servant David Faulkner:

> "Government regularly uses images and terminology of confrontation and warfare, with 'criminals' as an implied enemy who is of less value than the 'law-abiding' and 'hard-working' citizen .... Such language can also be heard as an encouragement or justification of abuses of power and due process. Its effect can be to deepen social divisions and increase the anxiety which the government itself wishes to prevent." (Faulkner 2014)

It is evident from recent ministerial comments that few lessons have been learned. If anything, the preference for inflammatory, exclusionary rhetoric has increased. Baroness Neville-Rolfe recently suggested that the Government should purchase its own planes to make deportation easier and cheaper and the Home Secretary has publicly blamed 'activist lawyers' for obstructing the deportation process (Hyde 2020). It seems probable that officials will continue to make the same mistakes when blinded by removal targets and a Government mantra that problematises all immigration (O'Nions 2020). As the UN Rapporteur on Poverty and Human Rights, Philip Alston, recognised in 2018, the 'hostile environment' goes to the heart of what it means to be British:

> "I wish to underscore that a hostile environment ostensibly created for, and formally restricted to, irregular immigrants is, in effect, a hostile environment for all racial and ethnic communities and individuals in the UK. This is because ethnicity continues to be deployed in the public and private sector as a proxy for legal immigration status. Even where private individuals and civil servants may wish to distinguish among different immigration statuses, many likely are confused among the various categories and thus err on the side of excluding all but those who can easily and immediately prove their Britishness or whose white identity confer upon them presumed Britishness" (UN Human Rights Commission 2018)

## 4. The Case against Deportation

Liability to deportation is one of a small number of provisions distinguishing citizens from non-citizens. As such it constructs citizenship, asserting its value as the highest level of belonging (Walters 2002, p. 288). Gibney describes citizenship as Janus-faced, as both a unifier that stresses a common identity and a divider that excludes non-members (Gibney 2011, p. 41). Every deportation affirms the significance of the unconditional rights of residence that citizenship provides whilst also affirming its normative qualities (Anderson et al. 2011). It can therefore be argued that deportation is constitutive of citizenship (Walters 2002, p. 288). This may go some way to explaining the increased use of citizenship deprivation powers and deportation since the UK voted to leave the European Union in 2016.

### 4.1. Constituting Britishness

Brexit has been inextricably tied to notions of identity to the extent that the enormous challenges of ensuring an orderly exit and the economic impact of leaving without a trade deal have been sidelined in the public discourse. Those that oppose Brexit are frequently depicted as destroyers of the democratic process, moderate 'remainer' conservatives were ousted from long-standing cabinet positions, leaving the country with an inexperienced government at a crucial time. The previous speaker of the House of Commons, conservative member of parliament John Bercow, was strongly criticized by the media and leave-supporting parliamentarians for asserting the power of parliament after Prime Minister Boris Johnson prorogued parliament in an attempt to reduce its scrutiny of Brexit legislation. Businesswoman Gina Miller who attempted to challenge the legality of the Brexit process in the courts found herself repeatedly depicted as an enemy of the people. Her dual nationality attracted particular condemnation; she was routinely described by tabloid newspapers as a 'foreign born multi-millionaire'.

Boris Johnson learned the lessons from his predecessor and rewarded loyalty above experience and proven competence. However, as the promise of an easy trade seems to be slipping away, Brexit-supporting MPs repeatedly downplay the economic arguments and stress that leaving the EU is about restoring independence from Brussels and securing British values. The nature of these values and their distinction from European values is impossible to pinpoint. However, the same values are commonly emphasised in public discourse over national security and public safety, particularly when the perceived threat comes from supposedly 'foreign' sources. The increasing danger posed by far-right extremists who have been galvanized by the Brexit process receives comparatively little media and political attention. Counter demonstrations to the 'black lives matter' campaign in several UK cities revealed an uncomfortable unity of perspectives, all of which mask deep-seated hostility

towards democratic values of equality, tolerance and human rights. The Government's attempt to unite the newly independent nation appears facile in the face of such polarized opinion. Reported hate crimes increased significantly after the referendum, even when compared to the number recorded following terrorist attacks in Manchester and London (Devine 2018). Devine concludes that media coverage of immigration has played a fundamental role in connecting 'meaningful democratic events' with 'prejudicial violence'. The most recent report of the UN Committee on the Elimination of Racial Discrimination described the referendum campaign as "marked by divisive, anti-immigrant and xenophobic rhetoric" which politicians failed to condemn, resulting in the creation and entrenchment of prejudices "thereby emboldening individuals to carry out acts of intimidation and hate towards ethnic or ethno-religious minority communities and people who are visibly different" (UN Committee on the Elimination of All Forms of Racial Discrimination 2016, para. 15). This context is very relevant to the public construction of the 'foreign criminal' as it allows a seemingly uncontroversial alignment of public safety and immigration control whilst affirming the normative value of citizenship.

## 4.2. Acquisition of Citizenship

British citizenship can be acquired through naturalisation which broadly depends on continued residence, evidence of integration and good character (required for almost all applicants over the age of ten). The UK's liberal nationalist model requires English language competence and completion of the 'Life in the UK' test. Naturalisation is then confirmed by an official ceremony and allegiance pledge. However, both the Life in the UK test and language competence are also required for indefinite leave to remain (ILR) applications, making the distinction far less significant.

Naturalisation requires a period of five years lawful residence (reduced to three in the case of spouses) with an additional one year spent with ILR. The rules for acquiring ILR as a child are easier to satisfy but the cost of then obtaining citizenship as an adult (currently £1032), along with the requirement of good character and the Life in the UK test, may be prohibitive for those from more disadvantaged backgrounds. It is quite possible that a person who arrived as a child with their family, like many of those recently deported, believes themselves to be a British citizen. The distinction is further blurred as commonwealth citizens who are ordinarily resident are able to vote in UK elections.

Although the number of people applying for citizenship has been broadly stable for the last four years it is less than half the figure for 2013 (Home Office 2018). Thus, many people remain in the UK and never acquire formal citizenship. Kanstroom (2007) describes them as 'eternal guests'. This absence of formal status has little significance unless the individual resides overseas for more than two years (in which case they will be subject to the returning resident rule) or if they engage in criminal behaviour. For those present without permission there is a requirement of twenty years residence to obtain leave to remain which may subsequently result in an application for ILR after an additional ten years. Naturalisation is therefore a distant dream for those with periods of unauthorised residence in their immigration history.

Those with a criminal record or previous immigration problems may find that citizenship is unattainable. Here we are reminded of the intersection between criminal and immigration law, what Stumpf describes as 'crimmigration law' (Stumpf 2006). Essentially this constitutes an additional range of sanctions only applied to the non-citizen, described by Bosworth as a kind of 'double jeopardy for non-nationals' (Bosworth 2011, p. 592). Stumpf likens the government's position to that of a bouncer whereby, upon discovering Kanstroom's 'eternal guest' is not a full member, there is enormous discretion to use persuasion or force to remove them from the premises (Stumpf 2006, p. 402).

## 4.3. Grounding the Rights of 'Eternal Guests'

There is a great deal of academic debate exploring the foundational principles of citizenship, in particular exploring the rights of 'eternal guests' when compared with formal members. Shachar is critical of the birth-right citizenship model, noting how heredity is rejected in almost every other sphere as a legitimate basis of discrimination yet in this context it is accepted as a just basis for the

distribution of additional rights and privileges. There is, she argues, a 'democratic legitimacy gap' when rights of equivalence are denied to long term residents (Shachar 2003, p. 347; Shachar 2009). Certainly, the coupling of fundamental rights with citizenship finds no place in international human rights norms, yet, as the Brexit process illustrates very well, there is a temptation for governments to exploit the line between citizens and foreign nationals in times of crisis. It is essential that fundamental rights and basic protections are grounded in human rights stemming from our common humanity. They should not be reframed as privileges dependent on citizenship (Cole 2006, p. 2543). Nevertheless, there is an evident tension between our conceptions of universal, inalienable human rights and the bounded nature of the modern democratic state. The resident foreigner is the incarnation of this tension (Gibney 2011, p. 45).

To the extent that there is consensus within the inter-disciplinary arena of citizenship studies it occurs when scholars address the rights of permanent residents who have built their lives in the country of residence (rather than nationality) (Young 2000). Whether their membership stems from vulnerability to state coercion, their contributions in the form of duties and taxes, or established societal ties, there are few scholars arguing that permanent residents should be treated less favourably than full members.

Miller argues that it is anomalous for someone whose interests are deeply impacted by the policies and laws of a state to have no say in determining these policies (Miller 2016; Walzer 1983). Baubock's stakeholder principle calls for an alignment between the reality of people's daily existence and their level of integration into society with their legal status (Baubock 2005, p. 667). But as Gibney notes, there are unanswered questions over how such integration can be measured (Gibney 2011, p. 66). This is particularly obvious when the individual engages in criminality. Whilst they may have a certificate confirming integration, their criminal conduct suggests otherwise and bars their movement to formal membership. Scholars are notably more cautious when advocating full membership in these cases. For example, Miller appears to accept the public interest argument that offending can legitimately lead to expulsion if the national community so determine providing basic procedural rights are protected (Miller 2016, p. 108).

Whilst Miller has been criticised for his defence of the status quo (Sager 2016), there are few who defend the membership of serious or persistent offenders. In this respect, many of the arguments applying rights to long-term residents based on their assumed membership centre on the figure of the sympathetic 'good migrant' whose presence is uncontroversial. These are relatively comfortable academic positions that avoid engaging with the most contentious issues of belonging and in so doing, it is suggested, they add weight to the position that the absence of a passport makes the offender inherently more dangerous.

Jospeh Carens rejects the conditionality of membership. In *The Ethics of Immigration* Carens persuasively argues that deportation of permanent residents is morally wrong for three interrelated reasons: membership, fairness to other societies and the rights of family members (Carens 2013, p. 102). It is the first reason that is perhaps the most contentious as it is unclear at exactly what point a person becomes a member. If, for example, a person has been in prison for most of their adult life will this prevent their membership? In the case of *Akinyemi No 2*, Judge Kecik took this approach in ruling that a thirty-three-year-old man was not socially and culturally integrated because he had a string of criminal convictions. The man in question had been born in the UK and had never left. The appellant *Kiarie* found that seventeen years living in the UK (since the age of three) was insufficient evidence for the Home Office to consider him culturally and socially integrated. Thus societal ties may be relatively easily denied when the individual concerned has a history of offending.

The question remains as to how membership should be measured, with the arguments appearing circular or arbitrary. There is clearly a distinction between the membership of non-citizens born in a particular country and those who arrive as adults or come for a specific purpose such as study or work. However, the social and cultural ties argument is not particularly helpful as this depends on the sociability and resources of the individual. The worker or student may actually accumulate more

social ties than the unemployed long-term resident, yet most would argue they have a weaker case for full membership.

Birnie builds on Carens' approach but seeks to avoid the limitations arising from a membership theory based on demonstrable social ties or contributions. His domicile approach stresses the importance of place which forms the backgrounds to our relationships and attachments to the natural and built environment and the social, economic and cultural activities that take place within it (Birnie 2020, p. 378). Birnie's theory is grounded in spatial rather than societal ties and it can therefore be applied to societal outliers, such as the hermit and the offender. It is also an individual rather than membership-based right that reflects the metaphysical nature of belonging and the importance this has for human flourishing.

Moore also advances a 'moral right of residency' that comes from the occupation of a space where we develop projects and relationships and pursue our way of life to which we are typically attached (Moore 2015, p. 38). Our space is fundamental to both the preservation of our life plans and continuing projects, but it also provides a deep emotional attachment to the place and the people therein. Whilst it is possible to have this connection to more than one place, Birnie argues this is relatively unlikely:

> "After a long period of absence from the country of origin of previous domicile, someone's geographically located projects and attachments there are by definition strongly diminished" (Birnie 2020, p. 380)

Importantly Birnie and Moore's positions avoid the limitations of a more subjective social ties approach which can be used to exclude those who engage in criminality or are perceived to live an isolated life (Stiltz 2013, p. 341). The place of domicile exists prior to and independent of the political community, what Walzer describes as a 'locational right' (Walzer 1983, p. 43). Birnie is clear that non-citizens who are effectively domiciled should have the same protection against deportation as citizens in order to "protect the integrity of their geographically grounded life project, regardless of whether they choose to naturalise" (Birnie 2020, p. 383).

This position reflects the lived reality of those deemed liable for deportation. They have typically spent their formative years in the UK and established relationships with families and friends to varying degrees. They may have worked or studied here but ultimately what connects them to the UK is a less tangible sense of place and home. In such circumstances, banishment appears particularly cruel and disproportionate.

It may be countered that the extension of unconditional rights of residence to those without formal citizenship would blur an already muddied distinction and would diminish the desirability of citizenship. Carens argues that naturalisation should not be required for the protection of rights of permanent residents. The option to freely consent to naturalisation is not always available but even if it were, inaction should not be used to justify the forfeiture of such vital interests:

> "If people are to give up a fundamental right, like the right to a live in a society in which they are most deep-rooted, it must be done as a deliberate and conscious choice" (Carens 2013, p. 103)

Whilst this may suggest that there is no substantive difference between social membership and full citizenship, Carens preserves some crucial distinctions. He argues that membership rights can be lost if the non-citizen voluntarily leaves the territory to reside elsewhere. He also draws an important distinction between the civil community and the political community of citizens that would exclude permanent residents who have not taken the final step of naturalising (Carens 2013, p. 102). The latter may enjoy additional privileges such as the right to vote and stand for office and this would preserve the ultimate membership status of citizenship. Birnie also preserves a distinction between citizenship and non-deportability, arguing that naturalization would protect a person who seeks to reside outside the citizenship state, giving them a right to return, whereas a domicile predicated right would be lost if the individual chose to relocate.

## 5. The Legal Power of 'Banishment'

British citizens cannot be deported unless their citizenship is first revoked, a power that has been increasingly used since the introduction of the hostile environment, but which is still largely confined to cases of suspected terrorism. There is a deep hostility towards non-citizens who commit crime as they are perceived to have abused the state's hospitality (Gibney 2013, p. 218). But the hospitality argument does not adequately explain why deportation is appropriate in the case of permanent residents who have acquired indefinite leave to remain. They have no immigration restrictions on their stay and regard the UK as their home. A deportation order typically begins with a period of detention which is subject to very limited safeguards and no maximum time limit (Bosworth 2011; Shaw 2018). The order remains in force for at least ten years until it is formally revoked; during this time the deportee cannot legally return. Banishment is therefore a better description of the deportation process.

The current law relating to the deportation of foreign criminals was introduced in 2007 following a scandal concerning the unsupervised release of an estimated 1000 foreign offenders. The scandal, described by Griffiths as a 'moral panic', led to the resignation of the Labour Home Secretary and the birth of the label 'foreign criminal' (Griffiths 2017, p. 530). Bosworth observes how "New Labour championed a rhetorical convergence between crime and immigration" focussed on public protection, the impact of which can be clearly seen today as migration and crime has become conflated in public discourse (Bosworth 2011, p. 587; Gibney 2013, p. 233). The expression of moral outrage over foreign criminals is clearly attractive from a political perspective, regardless of its efficacy in controlling immigration or reducing crime (Stumpf 2006, p. 413).

### 5.1. The Importance of a Label

The 'foreign criminal' label results in the complex intimate histories of a life being reduced to one defining moment. A study in Jamaica found that returnees struggled with the term 'deportee' due to its connotations ("no good, dutty (dirty) criminal") which hampered their efforts to meaningfully participate in society (Headley and Milovanovic 2016).

Having been punished for their lapse of judgement, the 'foreign criminal' continues to be labelled as a threat, entering a liminal state of deportability with the ultimate sanction being expulsion, a reminder that membership for the non-citizen is always contingent on good behaviour. Sigona's interviews with undocumented migrants, demonstrate how illegality permeates every aspect of life (Sigona 2012). Yet those who become 'foreign criminals' are not undocumented and have thus far not experienced this precarity. The offence changes everything. All other aspects of that person's life are trivialised as insignificant as Lady Stern highlighted with reference to Sakchai Makao who had been in the UK since the age of ten and faced deportation following an arson conviction:

> "he was not just a foreign national offender but a sportsman, a member of a family, a worker, a taxpayer, a member of a community and a constituent whose MP was very active on his behalf" (Gibney 2013, p. 232)

The 'foreign criminal' label obscures the richness and complexities of life with Mr Makao defined solely by this lapse of judgement.

### 5.2. The Introduction of 'Automatic' Deportation

Section 32 of the UK Borders Act 2007 provides for automatic deportation of 'foreign criminals' which becomes effective when the individual receives a prison sentence of at least 12 months. Further, it allows the Home Secretary to specify offences that are deemed to be 'particularly serious' where any sentence can constitute grounds for deportation. Regulations that set out offences, including

criminal damage, were declared unlawful by the Court of Appeal, resulting in a rebuttable presumption of dangerousness.[8]

Prior to the automatic deportation provisions, the Home Secretary could use discretion to deport and the courts could recommend deportation when sentencing, considering factors such as the nature of the offence, history of offending and assessment of risk. This therefore demanded specific consideration of the public interest. s117C of the Nationality Immigration and Asylum Act (hereafter NIAA) establishes that deportation is in the public interest. A foreign criminal may be detained immediately following the end of their sentence and there is no automatic bail hearing. They are typically detained for longer periods than other detainees, on average over four months (Shaw 2018, para. 4.98). The absence of a time-limit, the challenges of securing legal representation (legal aid is not routinely available in immigration cases) and anxiety over the ever-present possibility of expulsion leads many to report a deterioration in their mental health (Chief Inspector of Prisons 2017; Borril and Taylor 2009). One of the foreign criminals interviewed during Lord Shaw's investigation of immigration detention had been in the UK since birth. He had committed a gang-related offence as a teenager and was awaiting deportation to Nigeria, a country that refused to accept him. He had been detained at Campsfield House for more than fourteen months at the time of the report, notwithstanding a review recommending his release. His Home Office file again stated that he was not socially or culturally integrated in the UK due to his involvement in crime (Shaw 2018, para. 4.98).

Most legal systems allow opportunities for permanent residents to acquire citizenship and state practices preventing formal inclusion, such as the German Gastarbeiter system, typically attract criticism from those keen to ensure equal protection of the law (Castles 1985). The UK along with Denmark and Ireland have not opted into the European Directive on long-term residents 2003/109 which provides enhanced protection against arbitrary expulsion for third country nationals who are resident in a member state for five years. Article 12(1) states that an expulsion decision can only be taken where there is a sufficiently serious threat to public policy or public security and Article 12(3) requires that member states shall have regard to the duration of residence, the age of the person concerned, the consequences for the person and their family and links with the country of residence or absence of ties to country of nationality. The Court of Justice has confirmed that expulsions without consideration of these factors are unlawful even in cases where a person has received a custodial sentence.[9] (European Commission 2019).

Similarly, the UK government has not opted into Directive 2008/115 on Return of illegally present third country nationals, because it does not deliver a 'sufficiently strong returns regime' and is considered to be 'overly bureaucratic' (Nokes 2019). A more obvious problem conspicuously absent from Nokes's rationale is posed by Article 15 of the directive which sets out a six-month maximum period for immigration detention. Despite extensive domestic and European criticism, successive governments have refused to place a maximum time limit on immigration detention with the result that 12% are detained for longer than the European maximum. In 2018, 54 people were detained for longer than a year (House of Commons 2019). The UK is however bound by the Citizenship Directive 2004/38 which has been implemented through the Immigration (EEA) Regulations 2016. This requires that the removal of persons on public policy grounds who are exercising their Treaty rights of free movement requires an individual and present danger to one of the fundamental interests of society.[10] The current Home Secretary, Priti Patel, has recently announced that the UK Borders Act will be applied to EU nationals and their family members once the withdrawal period ends, whilst those who have received a one year custodial sentence will be banned from entering the UK.

---

[8]　EN (Serbia) v Secretary of State for the Home Dept.; Secretary of State for the Home Dept. v KC (South Africa) [2009] EWCA Civ 630.

[9]　Wilber López Pastuzano v Delegación del Gobierno en Navarra CJEU [2017] C-636/16.

[10]　See for example R v Bouchereau CJEU [1977] C-30/77.

### 6. Appealing against Banishment

Specific rights of appeal against deportation are contained within Part 13 of the immigration rules on the grounds of family and private life. Since the 2014 Immigration Act, these rules have been placed on a statutory footing by virtue of by s117C NIAA 2002. S117C states clearly that the public interest requires deportation and in the case of a sentence of at least four years the public interest requires deportation unless there are very compelling circumstances, over and above those described in Exceptions 1 and 2.

The exceptions, which are only applied to those sentenced below four years, centre on three scenarios.

(i)   A private life in the UK. This requires the appellant to demonstrate lawful residence in the UK for most of his life; social and cultural integration in the UK; **and** very significant obstacles to his integration into the country where they will be deported.

(ii)  A genuine and subsisting relationship with a qualifying partner, or

(iii) A genuine and subsisting parental relationship with a qualifying child, **and** the effect of the deportation on the partner or child would be unduly harsh. The qualifying partner needs to be British or have indefinite leave to remain and the qualifying child needs to be a British citizen or have spent seven years continuously in the UK.

The effect of the statutory underpinning is to ensure that both decision-makers and the judiciary have regard to the same criteria when assessing family and private life arguments. It can be viewed as an attempt to curtail judicial interference with executive decision-making. McCloskey J, President of the Upper Tribunal's Immigration and Asylum Chamber, referred to the statutory regime as 'novel and challenging', but which should be 'construed and applied in a manner which makes it sensible, intelligible and workable'.[11]

Questions have inevitably arisen over the significance of European Court of Human Rights (hereafter 'ECtHR') jurisprudence relating to Article 8 (family and private life) when considering deportation challenges. Regrettably, there has been little consistency in the judicial approach on Article 8. In *Hesham Ali* the Supreme Court had to consider the 'very compelling circumstances' test and applied a 'balance sheet' approach, reflecting Strasbourg jurisprudence and requiring a consideration of factors that are highly relevant to the social membership theory, whilst recognising that the public interest in deportation will almost always outweigh countervailing considerations of private or family life.[12]

Of particular relevance is the ECtHR jurisprudence requiring that special consideration be given to private lives formed when the deportee was a child, even in cases of persistent criminality.[13] In *Boultif v Switzerland* the ECtHR also had regard to the time elapsed since the commission of the offence and the appellants conduct following release.[14] Where a foreign criminal has not reoffended since their release this should refute a suggestion that they remain a threat to the public. This marks a recognition that the foreign criminal is not reducible to one moment in time. The absence of these considerations in s117 is deliberate. If the goal is automatic deportation of 'foreign criminals' there is no room for nuanced assessments of public risk.

In 2017 the Court of Appeal ruled that where there are no obvious compelling circumstances, they would not have regard to ECtHR jurisprudence and there was no judicial discretion to allow an appeal on human rights grounds.[15] This may appear to be a semantic exercise as where the 'very

---

[11]   Treebhawon and Others (NIAA 2002 Part 5A—compelling circumstances test) [2017] UKUT 13.
[12]   Hesham Ali v Secretary of State for the Home Dept. [2016] UKSC 60.
[13]   Maslov v Austria App. App 1638/03 23 June 2008.
[14]   Boultif v Switzerland App 54273/00 [2001] ECHR 497.
[15]   NE-A Nigeria [2017] EWCA Civ 239.

compelling circumstances' test is not met it is unlikely, given the public interest, that another human rights argument would prevail. However, it does provide an indication of the conflicted role of tribunals and courts who are being directed in how to undertake their judicial function on human rights assessments.

The case of *Akinyemi v SSHD (No 2)* is comparable to many of the more contentious deportation cases.[16] It concerned an appellant with a string of serious criminal convictions. He was 33 years old and had always lived in the UK. The Court of Appeal, applying *Hesham Ali*, noted that the public interest cannot be fixed in time. If deportation is to be compatible with Article 8 it must take into account:

> "such matters as the nature and seriousness of the crime, the risk of re-offending, and the success of rehabilitation, etc. These factors are relevant to an assessment of the extent to which deportation of a particular individual will further the legitimate aim of preventing crime and disorder, and thus, as pointed out by Lord Reed at para 26, inform the strength of the public interest in deportation"[17]

From this analysis one can perhaps be persuaded that the rights of offenders (and their families) are being fairly balanced against a carefully measured public interest, at least by the senior courts. However, few cases reach this level of judicial scrutiny and for those deported notwithstanding an arguable human rights appeal, the damage to family life is already being done.

### 6.1. 'Deport Now, Appeal Later'

The challenge of fighting a deportation decision is complicated by the non-suspensive appeal provisions introduced in 2014 as s.94B of the Nationality Immigration and Asylum Act 2002. The provision applies to human rights appeals and allows the Secretary of State to certify that removal would not violate s.6 of the Human Rights Act, for example because the individual would face 'serious irreversible harm'. The certification can be done after an appeal has been lodged and will then prevent the appeal from being continued while the appellant remains in the UK. It can be challenged by judicial review if notice is lodged within 5 days.

There are several issues with this approach. Underpinning all of them is a question over the appellant's safety on return. Jamaica is controversially included in the statutory list of safe countries where there is in general no serious risk of persecution (s94(4 NIAA 2002). It remains on the list notwithstanding a Supreme Court judgement in 2015 which ruled that where 10% of the population risked persecution on the grounds of their sexuality, there could not rationally be deemed to be 'no general risk of persecution'.[18] Until recently Jamaica had the highest homicide rate in the world, and it is still one of the most violent countries with a murder rate over forty times that of the UK and, according to Home Office figures, a 7% conviction rate (Home Office 2019a).

There are particular challenges faced by British deportees who lack resources and contacts to easily integrate. As Lord Shaw reported, most of those deported have no connection to Jamaica and have strong British accents making them clearly visible (Shaw 2018, para. 4.93). At least five British deportees are known to have been murdered in Jamaica since 2018 (Taylor 2019). Some of these cases relate to gang reprisals whilst others appear more random. Absent family, social and cultural ties mean it is difficult to imagine how a deportee can rebuild their lives without returning to criminal behaviour. Headley and Milovanovic (2016) suggest that deportees to the Caribbean are commonly blamed for the region's public safety troubles. This may be attributed to the US policy of deporting violent gang members to central and south America. However, it should be noted that an increase in deportations from the UK in the last twenty years often following conviction for drugs offences have contributed to

---

[16] Akinyemi No 2 v Secretary of State for the Home Dept. [2019] EWCA Civ 2098.
[17] Akinyemi No 2 v Secretary of State for the Home Dept. [2019] EWCA Civ 2098 per Lord Kerr, para. 49.
[18] R (Brown) v SSHD [2015] UKSC 8.

this association. Jamaican police have warned British expats that they face significant risks of financial crime and an 'extreme risk' of murder with at least 85 Britons, Americans and Canadians murdered between 2012 and 2018 (Halliday 2018).

Some have argued that safety considerations should be irrelevant when an offender is being deported. This is linked to the coupling of human rights with citizenship and the sense that the foreign criminal has abused their conditional membership. During the passage of the UK Borders Act David Davies MP argued that 'no country in the world be considered so dangerous that we should not deport people to it if they are persistent criminals or have committed serious crimes such as rape' (Davies 2007).

Davies also advanced a proposal that the age of liability for automatic deportation should be reduced from eighteen to sixteen (thus potentially including vulnerable children coerced into gang membership and drug dealing). Davies's arguments find favour with much of Britain's conservative media which repeatedly stress that 'foreign criminals' have forfeited their rights and are dangerous to the British way of life, whilst conveniently forgetting that they are British in almost every sense (see Drury 2020; Baker 2019).

Whilst the Home Office 2019 guidance recognises the severe pressures facing the criminal justice system in Jamaica, this receives little consideration prior to the deportation of those characterised as violent offenders who learned their craft in the UK. It will be recalled that 'fairness to other societies' was one of three reasons presented by Carens to support a moral right of membership (Carens 2013, p. 102). In his extensive review of immigration detention, Lord Shaw contests the presentation of all those deported as violent offenders, but he also goes further by questioning whether it is morally right to return any criminal whose offending follows an upbringing in the UK (Shaw 2018, para. 4.99).

## 6.2. The Effectiveness of an Appeal from Overseas

A second issue relates to the ability of the deportee to mount an effective appeal from overseas. This became the focus of the Supreme Court decision in *Kiarie and Byndloss* [2017] which appeared to sound the death knell for non-suspensive appeals.[19]

The appellants were from Kenya and Jamaica, respectively. Both had indefinite leave to remain and had established family lives in the UK. Mr Byndloss had a wife and four children and at least three children from other relationships in the UK. He was told that he did not have a subsisting relationship with any of his children. The Home Office rejected evidence from the prison records that his children had visited him during his incarceration. Mr. Kiarie was told that although he had been in the UK since the age of three, he was not socially and culturally integrated here and there would not be significant obstacles to his reintegration in Kenya. Their appeals against deportation following convictions for drugs offences were certified as clearly unfounded meaning that any right of substantive challenge would need to be made from overseas.

Giving the leading judgment, Lord Wilson confirmed that 'serious irreversible harm' may be caused to the individual and their family by separation, but he stressed that it could also result from an ineffective appeals process that undermines the right to appeal.[20] One of the central questions for their lordships was the extent to which an appeal from overseas could be a sufficient substitute for a UK tribunal hearing. The right to a fair trial in Article 6 of the European Convention on Human Rights does not apply to immigration proceedings as they are deemed to be administrative in nature.[21] However, it is now well-established that where the right to family and private life in Article 8 is engaged by a decision, that decision must carry with it the possibility of making an effective challenge.[22] In *Al-Nashif*

---

[19]　Kiarie and Byndloss v Secretary of State for the Home Dept. [2017] UKSC 42.
[20]　Ibid., para. 39.
[21]　Maaouia v France App 39652/98 5th Oct 2000.
[22]　R Gudadaviciene v Director of Legal aid Casework [2014EWCA Civ 1622 [2015] I WLR 2247.

*v Bulgaria* the ECtHR ruled that the refusal of a right to appeal where deportation interfered with the applicant's family life would mean that any such interference was not 'in accordance with the law'.[23]

In considering the effectiveness of remote appeals, Lord Wilson referred to Home Office statistics which suggested that an appeal would take a minimum of five months from overseas. In his opinion, this could significantly weaken the substance of the appeal and therefore it would necessitate considerable justification.[24] Appellants with limited means may also need to make an application for exceptional case funding under s10 of the Legal Aid, Sentencing and Punishment of Offenders Act 2012. This requires the appellant to show that the absence of legal aid *would* breach human rights, or it *might breach them and provision of it is appropriate in all circumstances.* Even if legal representation is secured the lawyer could face 'formidable difficulties' in giving and receiving instructions both prior to and during the hearing.[25] One of the biggest issues impacting on effectiveness is the ability of an appellant to give oral evidence. Given the appellant's character is so crucial to the success of an appeal, the provision of oral evidence and response to cross-examination is likely to have considerable impact.

Given the significant issues raised over the effectiveness of overseas appeals, it is perhaps worth considering why the Home Secretary introduced the certification process. It will be recalled that the focus of the 2014 Immigration Act was to reduce the number of appeals, thereby saving costs to the taxpayer, preventing abuse, and shoring up the integrity of the returns process. The specific focus of s. 94B was to reduce the delay in the determination of the appeal but also to prevent abuse by strengthening the ties of the deportee during the appeal process (May 2013a). The public association of appeal rights with procedural abuse has been a familiar theme in the rhetoric of the 'hostile environment' (see Hyde 2020; O'Nions 2020). The vital constitutional safeguard of judicial review has been presented as an abuse of the system at a time when the number of judicial reviews has fallen by 44% (Kate and Quinn 2020). In introducing the Immigration Bill before the House of Commons May stated:

> "Secondly, we will extend the number of non-suspensive appeals so that, where there is no risk of serious and irreversible harm, **we can deport first and hear appeals later**. We will also end **the abuse of article** 8. There are some who seem to think that the right to family life should always take precedence over public interest in immigration control and when deporting foreign criminals. The Bill will make the view of Parliament on the issue very clear." (May 2013b)

s.94B represents a pyric victory where costs to the deportee, their family and the taxpayer are likely to significantly exceed the cost of the system that it replaces. The cost to the taxpayer is difficult to determine as much depends on the specific facts and deportation may follow months of detention. Recent statistics show that 24,773 people were detained in 2018 and of these around 20% were actually deported, the majority of which were EU nationals (Home Office 2019b). Thus, the number of foreign criminals deported to non-EU countries is actually comparatively small and a significant number of those detained will be released back into the community (although their deportable status will remain). We do know that for the 46 people removed on six charter flights from July to September 2019, 203 guards were used and Mitie, who provide 'escorting' services, have a 10-year contract with the Home Office worth £525 million (Mitie 2017).

If the individual decides to purse their right to challenge their deportation, there will be more appeals and judicial reviews following the judgement in *Kiarie*. If the Tribunal concludes that the appellant needs to be in the UK to make an effective challenge, proceedings should be adjourned so that the appellant can return.

Given the complexity of the legal position and the costs associated with a protracted legal process, it is difficult to understand why the Government has maintained its position on non-suspensive

---

[23]  Al Nashif v Bulgaria [2003] 36 EHRR 123.
[24]  Kiarie and Byndloss v Secretary of State for the Home Dept. [2017] UKSC 42. para. 58.
[25]  Ibid., para. 60.

appeals. The answer may perhaps be explained by its dramatic impact on the number of appeals. In the eighteen months following its introduction the Home Secretary issued 1175 certificates pursuant to s. 94B in relation to foreign criminals, all, therefore, with arguable appeals. Of those the vast majority were deported in advance of their appeals but only 72 had filed notice of appeal with the tribunal from abroad. Not one of the 72 appeals had succeeded.[26] Given the badging of appeals as an 'abuse' of the system, one is led to conclude that this was more than an unforeseen consequence.

### 6.3. The Substance of Appeals

Whilst there are serious doubts concerning an appellant's ability to present a challenge from overseas, it must also be reiterated that certification implies that all those deported have arguable human rights cases.

The approach of Judge Kecik in *Akinyemi No2* illustrates the challenges faced by a deportee in demonstrating a private life in the UK when they have a history of offending. Yet if they are not deemed 'socially and culturally integrated' in the UK, it has to be concluded that they are not integrated anywhere. The relevance of offending to the degree of integration is problematic as the crime effectively becomes double-weighted. Whilst is clearly relevant to the strength of the public interest it now becomes relevant to the strength of the individuals' rights to a private life. Further, to conclude that criminality prevents social and cultural integration implies that British citizens, whose integration is a given, do not commit crimes; evidently a nonsensical conclusion.

There can be no doubt that the continued separation of families (including time spent in detention) will impact on a subsisting family life. But s55 Borders, Citizenship and Immigration Act 2009 gives effect to Article 3(1) UN Convention on the Rights of the Child by establishing that the child's best interests are a primary consideration in immigration cases. Applying this principle, cases such as *ZH Tanzania* [2011] have found that a mother's 'appalling immigration history' can be trumped by the best interests of her British citizen children. In the leading judgement, Baroness Hale emphasised that a child should not be blamed for the actions of her parents. In *Zambrano* the Court of Justice ruled that children who are citizens of member states have complementary Union citizenship which can prevent removal of an illegally present parent.[27] If removal of the parent would result in the child being compelled to leave the member state, the action will be unlawful.[28]

But the impact of the child's best interests in deportation cases is not so straightforward. The Supreme Court have reiterated that 'a' primary consideration does not elevate the child's best interests above all other considerations.[29] *ZH* is a removal rather than deportation case, so the public interest in expelling the parent is weaker as it centres on maintaining immigration control rather than public protection. The commission of a criminal offence strengthens the public interest considerably and the child's best interests may more easily be outweighed. The UK court have also reduced the impact of the child's interests when they are not British citizens (notwithstanding the absence of a citizenship requirement in the Article 3 of the Convention on the Rights of the Child). In *Zoumbas* the facts were comparable to *ZH* save for the absence of British citizenship.[30] The Supreme Court ruled that the parents with their three children could be removed to the Republic of Congo in the interest of maintaining effective immigration control.

For those with an established family life, the immigration status of their partner and children will therefore be relevant as is the need to demonstrate a subsisting relationship. This can be difficult if the appellant has spent time in prison and immigration detention. The impact on the family member

---

[26] Kiarie, para. 77.
[27] Zambrano (Gerardo Ruiz) v Office national de l'emploi CJEU [2011] C-34/09.
[28] Chavez-Vilchez and Others v Raad van Bestuur van de Sociale Verbekeringsbank and Others CJEU [2017] C-133/15; Patel, Shah & Bourouisa v Secretary of State for Home Dept. EWCA Civ 2028 [2017].
[29] ZH Tanzania v Secretary of State for the Home Dept. [2011] UKSC 4.
[30] Zoumbas v Secretary of State for the Home Dept. [2013] UKSC 74.

is assessed using the 'unduly harsh' test in the immigration rules. The Home Office defines unduly, according to the Oxford dictionary definition as 'excessively' and 'harsh' as 'severe or cruel' (Home Office 2019c). The guidance cites with approval the Supreme Court ruling in *KO Nigeria* that the 'unduly harsh' test is a high one, 'going beyond what would necessarily be involved for any child faced with the deportation of a parent'.[31] Authoritative guidance from the Upper Tribunal states that 'harsh' "denotes something severe, or bleak. It is the antithesis of pleasant or comfortable. Furthermore, the addition of the adverb 'unduly' raises an already elevated standard still higher."[32] If the child is not compelled to leave with the deported parent and has a good relationship with their other parent in the UK it will be particularly difficult to make this argument.

Cases, such as that of Mr Byndloss suggest that the Home Office may be routinely dismissing evidence of family life by placing an impossibly high threshold to determine that the relationship is 'subsisting.' This will become increasingly problematic when the family is separated by the non-suspensive appeal. When balanced against the public interest as defined in s117C NIAA there would seem to be very little opportunity for a foreign criminal to assert their fundamental rights before they are irrevocably damaged.

## 7. Conclusions

Detention and expulsion are not simply administrative acts to exclude undesirable immigrants. In the case of established residents, the 'domicile principle' should be applied such that removal is a disproportionate act constituting an additional punishment which is typically harsher than any imposed by the criminal justice system. This sentiment was captured by Justice Douglas in the US case of *Harisiades v Shaughnessy* in 1952:

> "If they are uprooted and sent to lands no longer known to them, no longer hospitable, they become displaced, homeless people condemned to bitterness and despair" (1952 cited by Schuck 2000, p. 67)

In his leading judgement in *Kiarie*, Lord Wilson acknowledged that the impact of removal on established family ties would 'probably be significantly more damaging than that of his prior incarceration here'.[33]

Liability to deportation leaves the 'foreign criminal' trapped in a state of perpetual quasi membership that can be withdrawn at any time. This should be conceptualized as a form of state tyranny (Walzer 1983, p. 62; Bosniak 2006; Carens 2013). The process of surveillance, further incarceration and deportation constitutes a substantial and enduring interference with the right to the private and family life of the deportee and their family members. The opportunity to rehabilitate and reintegrate following release from prison is not available to the 'foreign criminal' whose precarious status is confirmed from the point of first encounter with the police.

Prevailing human rights norms are decoupled from nationality and they should be sufficiently robust to defend the interests of all those subject to the state's jurisdiction. Yet, the margin of appreciation in the Strasbourg court has translated as judicial deference when it comes to public protection. Both decision-makers and courts appear reluctant to fully engage with the proportionality of deportation when the deportee is a 'foreign criminal' whose very existence is unequivocally presented as a threat to the public. Their family and private lives are devalued in the decision to deport and then purposefully undermined through a non-suspensive appeal process. Yet foreign criminals, their families and friends are also members of the public whose interests require protection by the state. The blanket public interest justification raises real ethical issues, obscuring the interwoven complexities of individual circumstances and personal histories, of lives made in Britain.

---

[31] KO (Nigeria) v Secretary of State for the Home Dept. [2018] UKSC 53.
[32] MK (Sierra Leone) v Secretary of State for the Home Dept. [2015] UKUT 223 (IAC), [2015] INLR 563.
[33] Kiarie, para. 58.

Those who have lawfully situated their lives in a state and thereafter established their home should be regarded as unconditional members of civil society. As such they should be immune from punitive 'crimmigration' measures. At present these measures are imposed as soon as the 'foreign criminal' completes their sentence. Release from prison starts a process of surveillance with the ever-present prospect of detention and expulsion, during which time migrants and their families "live in limbo where their lives are unsettled, ungrounded and uncertain" (Hasselberg 2015, p. 566). As recent cases illustrate, a short prison sentence is never spent for the 'foreign criminal' who may be detained pending deportation several years after release.

Once removed the 'foreign criminal' will struggle to access support networks and is likely to be viewed with hostility and suspicion in an unfamiliar, dangerous environment. In such cases, as Lord Shaw recognises, the deportee has little alternative but to return to a life of criminality (Shaw 2018, para. 4.95). This places an additional burden on the resources of the country of nationality. To the extent that any country is responsible for the conditions that contributed to the deportee's criminality it must surely be the country where they have spent their formative years.

In these circumstances, as Carens has argued, expulsion must be viewed as morally wrong from the perspective of membership, fairness to other societies and the rights of family members (Carens 2013, p. 102). It is also legally wrong for two principle reasons. Firstly, as critical assessments on individual circumstances and risk are side-lined in favour of blanket 'public interest' justifications. Independent judicial scrutiny of decision-making is undermined through strong statutory language in s117C NIAA that does not adequately reflect the jurisprudence of the ECtHR. Secondly, the ability to argue effectively against expulsion on human rights grounds has been deliberately eroded in such a way that it undermines constitutional protections.

Whilst there is unlikely to be significant public or political support for extending the rights of permanent residents to a position of near equivalence, much of the response depends on how these issues are represented. Following the Windrush scandal some of Britain's most anti-immigration newspapers recognised the injustice and highlighted many individual stories of hardship. The public comments on these stories are revealing. There is widespread sympathy centred around the Britishness of the Windrush victims, described in comments as 'one of our own' and 'citizens in all but name' (Tapsfield and Drury 2018). This suggests that the abolition of the hostile environment and its demonisation of all migrants and ethnic minorities is critical to a fairer model of membership that respects the fundamental rights of the whole community.

**Funding:** This research received no external funding.

**Conflicts of Interest:** The author declares no conflict of interest.

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
