# Peer review of "No Place Called Home. The Banishment of ‘Foreign Criminals’ in the Public Interest: A Wrong without Redress"

_laws, 2020_

Round 1

Reviewer 1 Report

This is a long and well-written paper. I have three main suggestions for improvement:

  1. The paper right now is clearly written for a British audience. There are references to Britain as "here," which are off-putting for readers in other places. There are also references to significant political figures and events in the UK that are not sufficiently explained for readers from other nations--for example, the Windrush scandal. If this paper is to effectively reach an audience outside the UK it should be revised with this goal in mind.
  2. The authors could take the counterarguments more seriously and follow the principle of charity, at least to an extent, in providing the strongest practical and philosophical arguments in favor of deportation. As it currently stands, the article seems to reject the other side very quickly, and to treat it dismissively (as "nonsensical," for example), when its position is mentioned at all. This undermines the argument being presented in the paper. If the best possible presentation of the opposing side were raised and considered, it would actually likely strengthen the argument being made.
  3. The paper moves back and forth between philosophical arguments and practical/legal details, such as the IT equipment used in remote courtrooms. It would be helpful to the reader to acknowledge that the paper is working on these different levels and explicitly address how the abstract/moral/philosophical interacts with the more applied arguments and concerns. Again, particularly for those reading from other countries, this would be especially useful.

Reviewer 2 Report

As I am in the midst of teaching a law course on the immigration consequences of criminal conduct in the United States, I was very eager to read your article. I was struck by the many similarities between the US and UK systems and the suspicion that the UK looked to some of the most draconian features of US immigration law as a model in its 2007 reform.

Overall, the article can be much more powerful with some organizational revisions and further unpacking of some of your arguments. The introduction begins with strength and clarity - that deportation as a consequence to criminal activity is ethically wrong based on notions of societal membership and legally wrong based on human rights norms. Yet you don't quite present those arguments and, in your final section (7), you seem to conclude that neither position is winning. (p14 "Prevailing human rights norms that are decoupled from nationality do not appear sufficiently robust to defend the right of the 'foreign criminal'"; p 15 "Whilst there is unlikely to be significant public or political support for extending the rights of permanent residents to a position of near equivalence...")

In section 4, you need to take ownership of your argument. You string together citations well, but it lacks the force of commitment. Make your argument!!! And if we agree with you that membership should control, what does this mean? Must the law be repealed? Or can/should the moral argument better inform the application of the law by the state or judges?

Similar focus is needed in sections 5-6. I assume this is your discussion of human rights legal standards, but those standards need to be raised up. Introduce them first (right to family? right of the child? I'm not even sure which ones you are considering.) and then review the UK legislation against those norms. You spend too much time discussing procedural issues that are not relevant to your overall argument. All you need to say about the appeals is that of 1175 decisions, only 72 appeals were filed from abroad and none succeeded.

I recommend moving section 5.2 on the 2007 Borders Act to section 2 in setting the context. 

Add explanations of some of the UK-specific references you make. For example, I do not understand what you mean by "victim of county lines drug trafficking" (p3), and I still don't quite understand whether "hostile environment" is just a generic description or if it carries some particular meaning. I was also not familiar with the Windrush cases or Operation Nexus or the "Lessons Learned" document. You use the term "here" a couple of times in the article presumably for the UK, but it is unclear. Better to just use "UK." Finally, pay attention to the language you use. In sections 1 and 2, you use the terms "criminals" and "foreign criminals" as a seeming endorsement. You might use non-citizens. Or if you are adopting statutory language, perhaps put it in quotes.

Reviewer 3 Report

GENERAL OVERVIEW

The introduction presents a brief overview of the context of the research mainly focused on the author findings. It is beneficial that the author anticipated the finding in the introduction. Yet, more information on the context would have been beneficial. As an example the author takes for granted the definition of foreign criminals that is specific to the UK legal system and could not find a similar notion in EU or in the jurisprudence of European Court of Human Rights. The trend towards criminalization of non-citizens could be found in other countries also as a reply to the immigration crisis. For this reason and as noted in more details in the final section of this review, the Reviewer would like to suggest to better include the context of the research and other countries experiences especially of EU on the topic. An outline of the project should be added at the end of the introduction.

The paper is logically organized. The author, after the introductory remarks, addresses the problem of foreign criminals with examples under section 2 then shows the changes in UK which followed the Windrush scandal in section 3 and 4. Section 5 is dedicated to the analysis of the legal power of banishment and 6 to the administrative remedies before the concluding remarks.

The conclusions are solid and appropriate mostly focusing on how to better the UK legal framework to strengthen foreign criminals and how to develop effective administrative remedies. Yet, more information on the relations between on one hand UK and EU return policies and on the other on Brexit could be beneficial to further strengthen the quality of the paper.

English language and style are appropriate however a final check is needed

The reviewer did not spot any noticeable mistakes. Language is appropriate for academic writing. Yet, some inconsistencies should be reviewed. The author states at line 229 ‘Similar concerns were raised five years ago..’ while the direct quotation is from 2004. Similarly, the author states at line 52 ‘Recent amendments..’ but failed to specify the year of the amendment. Further in the abstract at line 6 a full stop is missing. The reviewer would like to suggest a language revision to eliminate the abovementioned spotted inconsistencies.

Additional Comments and suggestions to the authors:

This article focuses on a timely and interesting topic the problem of foreign criminals and deportation. Since 9/11, laws and regulations have been focused on tools to strengthen the control of criminals especially the ones committing terrorism-related offences or crimes against the state. As correctly noted by the author, criminalization is still based on the divide amongst citizens and non-citizens or better holders of long term residence permit in the country. The problem is affected by a different degree of protection between the two groups in the latter case (foreigners) remedies against decision of banishment are scant and ineffective. The paper critically assesses how public interest is often instrumental used as a justification for the different treatment. The findings highlight how the UK system fails to guarantee rehabilitation and reintegration in the society of non-citizens and how this situation is affecting also family and people directly linked with the offender. The paper suggests to better acknowledge under the legal framework at lines 710-711 that those who have situated their lives in the UK and established their place and  domicile here should be regarded as unconditional members of civil society. The author addresses, even if only partially, the Brexit and how the current environment, also as a result of the rise of political populism, is unwillingly to support extension to non-citizens of the rights of citizens which are in fact the strongest remedies against the power of the state. While the analysis is solid and covers a really interesting and time sensitive topic, it would have been beneficial to better cover the relations between the EU legal framework and the UK and offers more information on the differences between the two systems. DIRECTIVE 2008/115/EC for the first time established a common framework for the problem addressed in the paper and the European Court of Justice case law clarified the aims of the directive by setting the foundation for EU immigration policies. By adding also the position of EU to the topic the concluding remarks and suggestions could be fortifies. On the topic the reviewer would like to suggest the reading of:

Basilien-Gainche, Marie-Laure. “Immigration Detention under the Return Directive: The CJEU Shadowed Lights.” European Journal of Migration and Law 17, no. 1 (2015): 104–126.

Canetta, Emanuela. “The EU Policy on Return of Illegally Staying Third-Country Nationals.” European Journal of Migration and Law 9, no. 4 (2007): 435–450. https://doi.org/10.1163/138836407X250490.

Desmond, Alan. “The Development of a Common EU Migration Policy and the Rights of Irregular Migrants: A Progress Narrative?” Human Rights Law Review 16, no. 2 (2016): 247–272. https://doi.org/10.1093/hrlr/ngw012.

Eisele, Katharina. Return Directive 2008/115/EC: European Implementation Assessment. Brussels: European Parliament, 2020. https://op.europa.eu/publication/manifestation_identifier/PUB_QA0320329ENN.

Hinterberger, Kevin Fredy. “An EU Regularization Directive. An Effective Solution to the Enforcement Deficit in Returning Irregularly Staying Migrants.” Maastricht Journal of European and Comparative Law 26, no. 6 (2019): 736–769. https://doi.org/10.1177/1023263x19866541.

Lutz, Fabian, and Sergo Mananashvili. “Return Directive 2008/115/EC.” In EU Immigration and Asylum Law, 659–764. Nomos Verlagsgesellschaft mbH & Co. KG, 2016.

Moraru, Madalina, Galina Cornelisse, and Philippe de Bruycker. Law and Judicial Dialogue on the Return of Irregular Migrants from the European Union. Modern Studies in European Law; ; Volume 99. Oxford [England], London, England]: Hart, Bloomsbury Publishing, 2020.

Niovi Vavoula. “The Interplay between EU Immigration Law and National Criminal Law: The Case of the Return Directive.” Research Handbooks in European Law Series. Edward Elgar Publishing, 2016. https://doi.org/10.4337/9781783473311.00025.

Obenius, Hedvig. “The Return Directive 2008/115 EC and the Concept of Cooperation,” 2014.

Progin-Theuerkauf, Sarah. “The EU Return Directive – Retour à La «case Départ»?” Sui-Generis.Ch, 2019. https://doi.org/10.21257/sg.91.

Trojanowska-Strzęboszewska, Monika. “The Impact of the Migration Crisis on the Process of Shaping Goals and Instruments of the EU Immigration Policy.” Modelling the New Europe. An On-Line Journal, no. 26 (2018): 169–190. https://doi.org/10.24193/OJMNE.2018.26.11.

Round 2

Reviewer 3 Report

The revised article is solid and ready for publication. The author improved the coverage of the essay by including dedicated paragraphs to the situation in EU countries and the rationales behind the UK decision to opt out from EU directives on migration. The reviewer agrees with the author that the article is already pretty long and adding additional information could be problematic. The reviewer also supports the author decision to develop in a new article the post-Brexit situation.